# Decentralised trials for hearing and tinnitus therapies: Lessons from the Digital thErapy For Improved tiNnitus carE (DEFINE) randomised controlled trial

Joseph Salem[1,2], Dhiraj Sharma[3,2], Amy Moore[4], Olivia Dantonio[4], Luke Twelves[4], Emma Ogburn[4], Michael Young[4], Jan Multmeier[4,5], Jameel Muzaffar[2,6], Matthew E. Smith[7,8]*

1 Univeristy College London Hospitals NHS Foundation Trust, London, United Kingdom, 2 Oto Health, London, United Kingdom, 3 Imperial College Healthcare NHS Foundation Trust, London, United Kingdom, 4 Lindus Health, London, United Kingdom, 5 TotalEffects, Berlin, Germany, 6 University Hospitals Birmingham NHS Foundation Trust, Birmingham, United Kingdom, 7 University of Cambridge, Cambridge, United Kingdom, 8 Cambridge University Hospitals NHS Foundation Trust, Cambridge, United Kingdom

* m.e.smith@doctors.org.uk, mes39@cam.ac.uk

## Abstract

Randomised Controlled Trials (RCTs) are the gold standard for evaluating the efficacy of interventions; yet, traditional methods involving multiple recruitment sites often involve significant logistical and financial challenges. The DEFINE trial demonstrates the feasibility of a decentralised approach to RCTs by comparing smartphone-delivered self-guided tinnitus therapy against one-to-one therapist-facilitated treatment for tinnitus. This trial was conducted entirely remotely, leveraging digital technologies for remote recruitment, data collection, and intervention delivery. A total of 210 participants were recruited through social media platforms over a five-month period. Participants were screened and enrolled by a central trial team remotely, who utilised hearing test smartphone applications and electronic consent forms. Baseline and follow-up assessments were conducted using electronic data capture (EDC) platforms, with high retention rates observed at each time point. The trial successfully recruited and retained participants, demonstrating the efficiency and cost-effectiveness of remotely managed trials. Key findings include a high engagement rate from social media ads, with 151,978 impressions leading to 4,997 clicks (3.3%), with a direct advertising spend of £880. 912 individuals self-screened for eligibility online. The median age of participants was 58.3 years, in line with comparable traditionally-recruiting tinnitus studies, with good geographical distribution across the UK. The trial's adaptability allowed for protocol adjustments, and real-time monitoring of data quality and completeness. The DEFINE trial demonstrates that decentralised RCTs can offer a viable alternative to traditional RCTs for some hearing and tinnitus research, potentially increasing participant diversity and

**Data availability statement:** Our submission contains all raw data required to replicate the results of the study.

**Funding:** This study was funded by INNOVATE UK Smart Grant (Grant No. 10062270) and Oto Health Ltd. internal funding. INNOVATE had/will have no role data collection and analysis, decision to publish, or preparation of the manuscript. Oto Health Ltd has been involved in study design, and will assist with preparation of manuscripts. Data collection and analysis are provided by an independent contract research organization (CRO).

**Competing interests:** Oto Health Ltd. provided internal funding for this study. JMuz is Chief Scientific Officer for Oto Health Ltd, JS is the Founder's Associate at Oto Health Ltd, and JS and JMuz have financial interests in Oto Health Ltd. This does not alter our adherence to PLOS ONE policies on sharing data and materials. There are no products in development associated with this research to declare but the Oto app is commercially marketed by Oto Health Ltd.

reducing the burden of research on participants, while maintaining rigorous standards of data collection and participant safety. Increasing clinical use of remote audiological assessment, and hearing implant programming provide increasing opportunities for the adoption of entirely remote or hybrid studies in hearing and tinnitus conditions.

## Introduction

Randomised controlled trials (RCTs) provide one of the highest levels of evidence for assessing the relationship between treatment and measured outcomes [1]. The design of an RCT can have a direct impact on the quality of evidence collected and the inferences made between treatment arms and their relationship with the outcomes collected. High quality RCTs require extensive planning for effective participant recruitment; safe, consistent interventions; compliant data storage methods; accurate and timely data collection and effective trial management, including communication pathways and contingency plans. Traditionally this has required a large budget secured from either public or private grant funding and long project runways for the submission of protocols and approvals to the relevant organisations and authorities. Delivery of RCTs is usually by centrally coordinated but distributed clinical research teams, usually based in universities or large research-intensive institutions. Such site-based RCTs have some benefits, including access to resources such as laboratories, experienced personnel and organised and protocoled communication hierarchies. However, such approaches are costly and slow to set up and can lead to geographic mismatches between populations with high disease prevalence and viable trial institutions, reducing the diversity of trial participants. Trials can also place a high burden on participants, often exacerbated by long travel times. It is essential that trials involve a representative sample of the target population, and many funders are exploring ways to diversify the cohort participating in research, taking research to those who most need the interventions [2].

The use of remote, usually digital approaches to recruitment, consent and data collection, in what are commonly termed decentralised clinical trials, has recently changed the way some RCTs can run, through reducing administrative burdens on trial organisers and streamlining processes for both time and resource efficiency. Examples of these include participant recruitment and selection, traditionally completed through advertising to patients via letters or posters in community centres or hospitals with face-to-face interviews to assess eligibility criteria. Now recruitment information can be disseminated via social media and internet links with eligibility interviews completed online, thereby increasing the breadth of patients reached and reducing time commitments for both prospective participants and recruiters. The geographic limitations of a physical Clinical Research Organisation (CRO) site are being relieved through the creation of CROs which can utilise a fully decentralised model to conduct all trial procedures remotely with digital support. This approach involves a central trial team of investigators, trial managers, trial nurses and coordinators, trained in good clinical practice and trial protocols, conducting the trial remotely.

Remote methods allow the team to maintain regular communication with patients irrespective of where patients are being enrolled, or researchers are based. Whilst effective hypothesis generation, sufficient study power, bias management and study costs and time restraints remain sizeable hurdles to overcome, digital technologies can streamline RCT setup, monitoring and data collection, providing the opportunity to deliver a larger number of studies across a wider variety of locations. Interest in this area led to the creation of National Institute for Health and Care Remote Methods of Trial Delivery guidance [3].

Tinnitus is the perception of sound in the absence of an external stimulus and affects approximately 10% to 15% of people in the UK with prevalence expected to grow with an ageing population and increases in noise-induced hearing loss [4]. For those severely affected, tinnitus can significantly reduce quality of life [5,6]. Current National Institute for Health and Care Excellence (NICE) guidelines recommend amplification devices and psychological therapies for severe tinnitus management, including cognitive behavioural therapy (CBT). In the UK resource limitations mean tinnitus therapy provision is varied, and not widely available within the National Health Service. These problems are exacerbated by the long waits for treatment with patients often waiting many months to see an Ear, Nose and Throat doctor, prior to a further wait to see an audiologist/hearing therapist to access therapy [7]. As a result, online CBT has been developed [8] to mitigate these issues and provide more accessible care to people with tinnitus, irrespective of their geographical location or position on a hospital waiting list.

We present our experience of the feasibility and delivery of a decentralised RCT for tinnitus in the Digital thErapy For Improved tiNnitus carE (DEFINE) trial, comparing smartphone delivered self-guided tinnitus therapy including CBT against one-to-one tinnitus therapy including CBT facilitated by a human therapist. This article will not present results, rather focussing on methodology and learning points for decentralised trials.

## Methods

The DEFINE trial was delivered entirely remotely, with no direct face-to-face participant contact, and no requirement for participants to leave their home. Participant contact and management was provided by a Clinical Research Organisation (Lindus Health Ltd). A full protocol was previously published [9] and the trial registered online (ISRCTN99577932). NHS ethical approval was granted by the West Midlands – Black Country Research Ethics Committee (REC Reference: 23/WM/0146) with all participants providing written consent to be involved in the study.

### Recruitment

Participants were recruited using social media advertising to identify potential participants experiencing intrusive tinnitus. Adverts, such as the example shown in Fig 1, were created using ethics committee approved language and images and posted to Facebook and Instagram channels. Standard advertising algorithms were used to target those with an internet search history related to tinnitus.

Interested and potentially eligible participants were directed to the trial web page which outlined key information about the trial, including information about the Oto Tinnitus Programme, cognitive behavioural therapy for tinnitus and the aims of the trial. This is shown in Fig 2. The webpage also informed potentially interested participants of the study timeline (12 months), key eligibility criteria and the target number of participants.

### Participants

Potentially interested participants were directed via the advert to complete an online pre-screening form to assess their eligibility for the trial. The pre-screener included the following eligibility items:

• Age: 18 years or older

• Tinnitus symptoms that have lasted at least 3 months

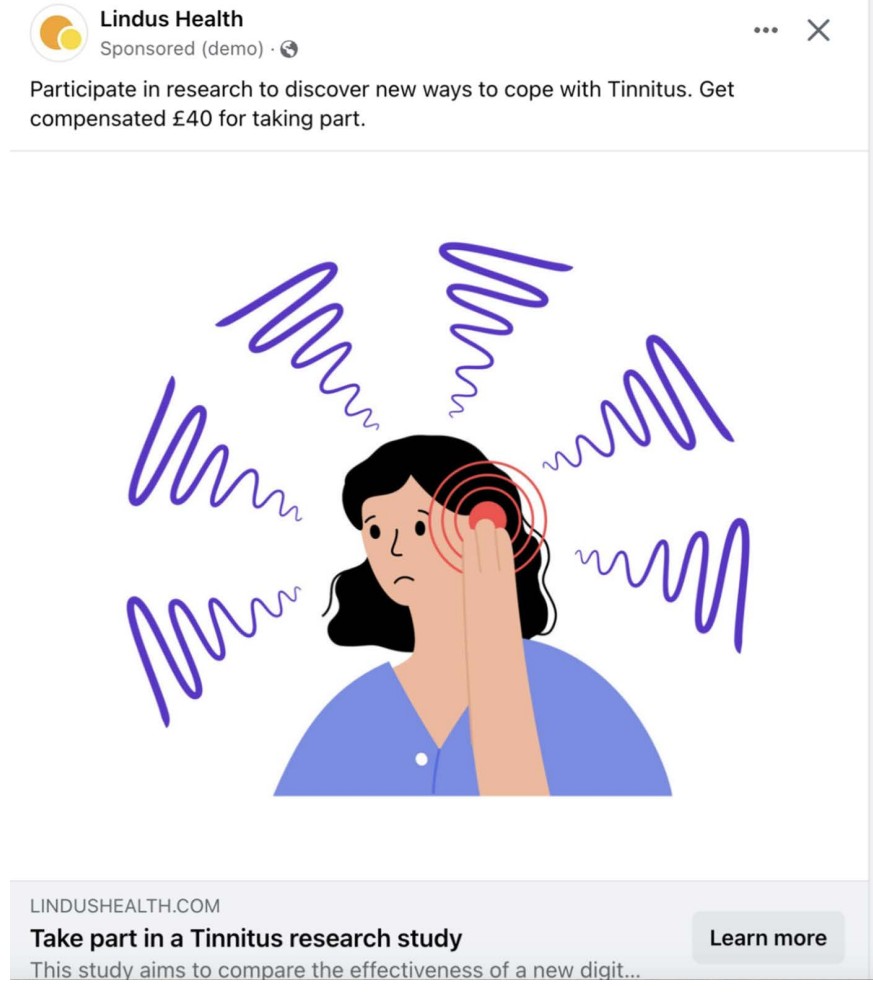

**Fig 1. Sample social media advertisement.**

- Tinnitus symptoms frequently and significantly impacting quality of life

- If the participant had received support to help with tinnitus, e.g., CBT

- If the participant had a smartphone with headphones

## Eligibility and consent

Once the pre-screener tool was submitted, participants who were potentially eligible were sent the Patient Information Sheet (PIS) and Informed Consent Form (ICF), as well as a link to an animated video explaining the trial. The participant was also invited to book a telephone/video call with a research nurse in the central trial team, using a web-based timetabling application (Calendly). Participants were asked to have headphones available at the time of their call.

The participant attended a video call with the trial team with various time slots, including the morning, afternoon and early evening to ensure flexibility to accommodate participant's schedules. Reminders were sent via email and text message to support attendance. Given the low risk of intervention, participant eligibility was assessed by a member of the

### The DEFINE Trial – Digital Therapy For Improved Tinnitus Care

Tinnitus therapy, including Cognitive Behavioural Therapy (CBT) is an effective treatment for helping people to manage tinnitus. This study aims to compare the effectiveness of a new digital tinnitus therapy, The Oto Tinnitus Programme, to conventional tinnitus therapy.

The Oto Tinnitus Programme is a digital approach to tinnitus management that delivers therapies through a self-paced smartphone app.

The DEFINE study will randomly assign adults with tinnitus to receive the Oto programme or therapist-delivered conventional tinnitus therapy, evaluating both their effectiveness in reducing tinnitus severity as well as the financial implications of both approaches.

Participants will be required to complete electronic surveys throughout the 12-month study period while receiving therapy for their tinnitus. You will be compensated £40 for your time.

Click here to take part

**Fig 2. Define Trial website landing page.**

central trial team who assessed this against the self-reported medical criteria to determine suitability for the study. Electronic consent via Docusign (Docusign Inc, CA, USA) enabled participants to provide consent without having to attend for an in-person visit. A record of screen failures that did not meet the inclusion/exclusion criteria was retained.

Participants were directed to download a free smartphone application to screen their hearing for the sole purpose of stratification at randomisation ('Easy Hearing Test' application for iOS devices (Hiroaki Ito) or 'Hearing test' app for Android devices, (e-audiologica). Neither app required sharing of personal identifiable data, nor had been previously validated in the home environment [10]. Participants were instructed on each app and allowed to complete the hearing test for both ears, typically taking around 5 minutes. They shared the audiogram result for screen capture and central database storage. All participants were encouraged to seek formal audiometric testing rather than to rely on the screening hearing test as it was made clear that these results are not diagnostic.

### Interventions

Once enrolled, participants were assigned through a stratified 1:1 randomisation by computer algorithm (permitting minimisation for several variables, including hearing level) to one of two arms: the intervention, the Oto Tinnitus Programme smartphone application (Oto Health Ltd, London, UK), or the control, one-to-one therapist-delivered tinnitus therapy. Both were delivered fully remotely.

If allocated to the Oto Tinnitus Programme, the participant downloaded and registered the Oto app. They could then progress through the programme, which incorporates elements of education, CBT, mindfulness and sound therapy. If allocated to the standard therapy arm the participant booked an online video appointment with one of the trial therapists. Appointments were available within and outside working hours for flexibility. Supplementary Information shows the Control Arm Intervention Specification. The Trial flow can be visualised in Fig 3.

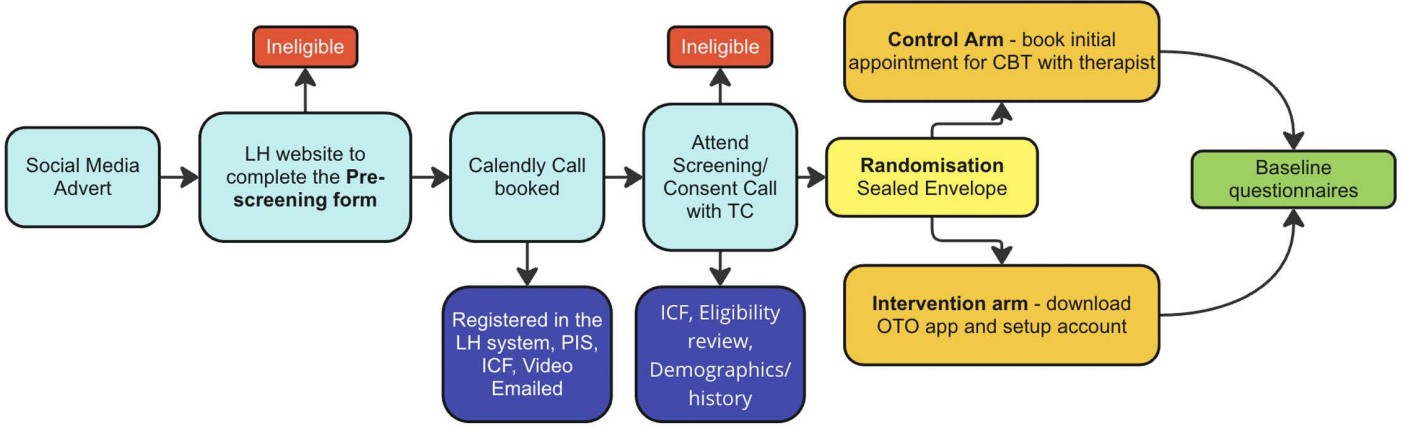

**Fig 3. Trial flow.**

## Data collection

The primary outcome was the change from baseline in Tinnitus Functional Index (TFI) at 6 months. The TFI is a well validated disease-specific patient reported outcome measure (PROM) [11]. Other PROMs included the HUI and EQ5D-5L [12,13], and adverse event data were collected. Resource use questions were additionally included for health economic analysis, as were usability and experiences questions for the interventions. Participants were required to complete questionnaires at baseline, and 1, 3, 6 and 12 months after randomisation.

Participants completed the questionnaires directly into the Lindus Health Electronic Data Capture (EDC) platform, via links sent via email and/or text.

Data completeness was enhanced via several approaches. For patient reported outcomes, the EDC platform notified both the participant and the study team if assessments were overdue. Missing data triggered an alert message to the participant via text/email (according to participant preference). If no response was entered, a daily reminder would be triggered for 3 days following the initial message, providing a link to the survey requiring completion. Trial coordinators could contact the participant up to three times via email, text and phone. Participants were also incentivised with shopping vouchers issued for all outcome assessments completed (baseline £10, Month 1 £5, Month3 £5, Month 6 £10, Month 12 £10).

Intervention adherence was monitored via app use data, and using reports issued by tinnitus therapists.

## Qualitative interviews

A sample of participants who consented to being further contacted were invited via email to join semi-structured interviews at three months or focus groups at six months after randomisation (following primary outcome data collection). Those potentially interested booked an information and consent call with the trial coordinator. Interviews and focus groups were scheduled on weekday evenings to accommodate the maximum number of participant schedules and conducted by a trained facilitator. Discussion based on an approved Topic Guide focussed on the participant's experience with tinnitus as well as their feelings on the content and delivery of the intervention.

## Results and discussion

The DEFINE trial concluded in December 2024, with data analysis ongoing. Initial findings provide evidence of the benefits of the decentralised model for study delivery, particularly in the field of tinnitus research and in the assessment of

digital therapeutics. Below we discuss the experience of delivering DEFINE in relation to the benefits and limitations of a fully remote, decentralised approach.

## Data completeness and participant retention

All assessments were completed online. Notably, 74% of participants completed the baseline surveys on the same day they received the link, and an additional 14% did so within one day. The remaining participants completed between two and five days after receiving the survey. Participants received a daily reminder for three days following the initial message, 24 hours apart, typically consisting of two email reminders and a final phone call reminder on Day 3.

For the Month 3 assessments, 77% of participants completed the surveys on the day the survey was sent out, with a further 15% completing it one day later, following the first daily email reminder. At Month 6, the primary endpoint, 80% of participants completed the surveys within one day. The remaining 20% completed the survey within two and five days. To increase retention at the Month 6 timepoint, the second reminder was adapted from email to SMS, which may be linked to a slight increase in survey completion on day 3.

After accounting for withdrawals, the proportion of participants submitting outcome data in DEFINE was high; 95%, 95%, 93% and 97% at 1, 3, 6 and 12 months respectively (Fig 4). These figures are favourable compared to similar trials [14] and comparable to rates in most conventionally-delivered trials. Decentralised trials typically use an electronic data capture tool, which can be supplemented if required with paper forms returned directly from participants. As found in DEFINE, this ensures no loss of data between sites and CRO, central access to source data and good accuracy without transcribing errors.

## Effective participant recruitment

The DEFINE trial social media campaign launched on July 25, 2023. In the first week of live adverts, 166 people completed the pre-screener form. Of these people, 130 were pre-eligible based on their responses. In total, social media adverts received 151,978 impressions and 4997 clicks – i.e., 3.3% of those shown an advert chose to learn more.

912 potentially eligible participants completed the pre-screener over the six-month period. Of these, 558 were eligible based on the pre-screening items for a pre-screen success rate of 61.1%. A further 452 potentially eligible participants (81%) booked a call to complete screening and eligibility with the trial coordinator. All participants successfully shared the screening audiogram result. The participant numbers at each stage of the trial are shown in the CONSORT diagram, Fig 4. In total, the trial enrolled and randomised 210 participants over a period of 5 months, with monthly figures noted in Table 1.

Conventional site-based trials are frequently slow to open sites due to contract negotiations, and once open, identification of participants can take time, particularly given the limited patient pool for any single hospital or health centre. The DEFINE trial recruited the first participant within seven days of the initial advert and successfully recruited all participants over just five months. This success arose from being able to reach a national audience via social media, with the ability to direct advertising to demographics more likely to have the target medical condition. Recruitment was so successful that the social media campaign needed to be paused to allow for participant processing, accounting for the drop in recruitment numbers in September.

## Enhanced participant reach

All trials will include population biases, but well-designed studies with effective recruitment strategies can reduce this effect. As a starting point, a decentralised trial should not introduce excessive bias into the participant population, and ideally enhances diversity.

Of those recruited, 47% were male and 53% female. The median age was 58.3 years, with the 60–69-year age group most represented (Fig 5). These figures are comparable to other tinnitus trials that recruited from healthcare sites in a

DNA = Did Not Attend, LTFU = Lost To Follow Up.

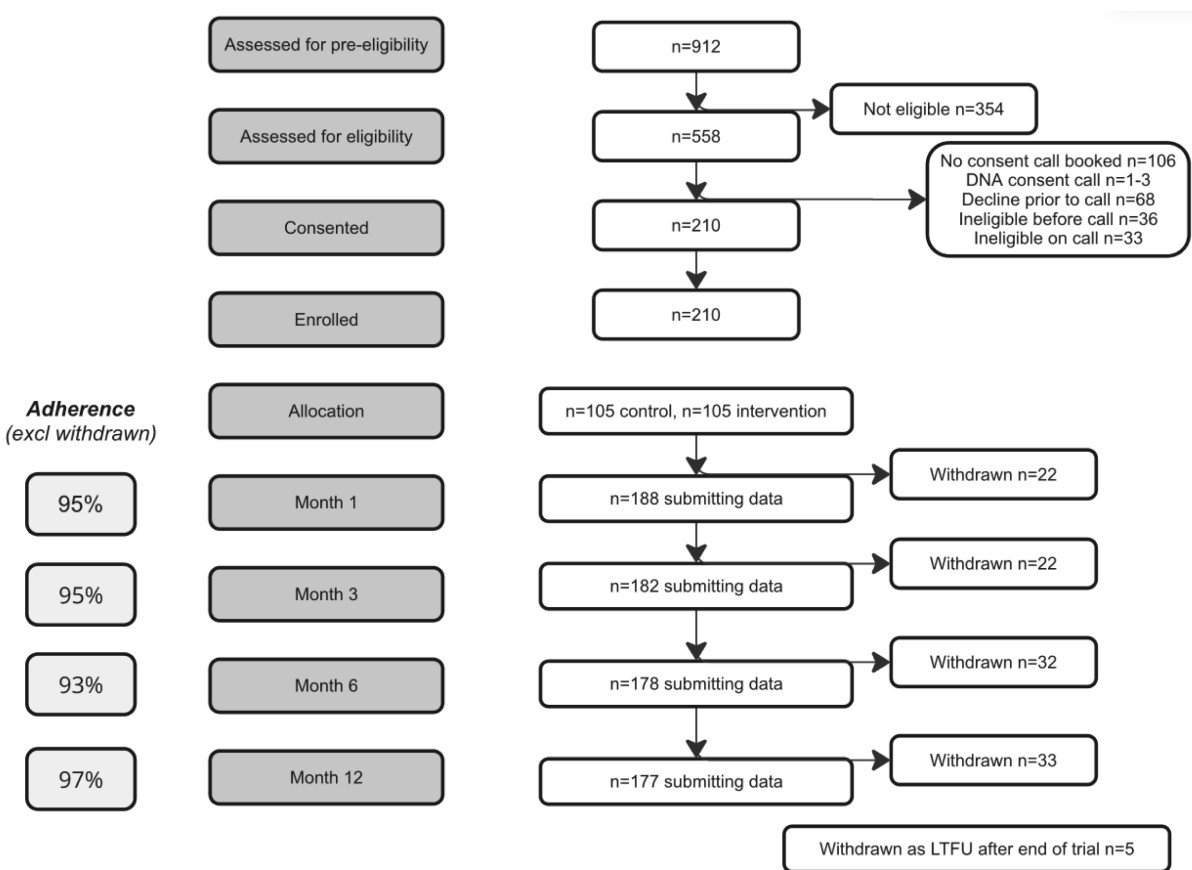

**Fig 4. DEFINE trial CONSORT diagram.**

conventional trial design [15] and studies using advertising via paper print and other media [8]. Of note, the digital nature of the interventions and trial advertising did not skew demographics to a younger cohort.

DEFINE also achieved a broad geographic distribution of participants across the UK (Fig 6). To reach the geographic mix of patients seen in DEFINE using a conventional recruitment structure would have required a very large number of sites. Trial designs such as DEFINE could go some way to reach populations not historically included in research, for example people in full time employment, those caring for young children or people living in remote areas. Although many

**Table 1. Recruitment metrics August-December 2023.**

| Month | Number of pts. recruited |
|---|---|
| August | 72 |
| September | 28 |
| October | 77 |
| November | 27 |
| December | 6 |

demographic variables were not collected to validate the inclusion of these groups, participants feedback and qualitative interviews identified that those involved valued the flexible timing of therapy appointments and data entry, and the lack of travel requirements.

It must be acknowledged that social media advertising, particularly using only certain platforms, may have introduced a demographic bias in the trial cohort. Although age bias is not clearly seen, it is possible that participants are more likely to be English speaking, educated and socially active, with the requirement for connected technology possibly excluding those who are unemployed or on low incomes. Given that access to technology was also required for participants to complete the study interventions, this bias is also relevant when looking at implementing these technologies into healthcare.

## Reduced patient burden and increased retention

Conventional clinical trial designs often centre around patients travelling to a local trial site for interventions and assessments. Decentralised trials such as DEFINE remove the need for trial sites, introducing remote methods of study delivery. Travel time is an important consideration for participants, with distance adversely impacting trial participation and completion [16].

One risk of the decentralised trial methodology is that patients feel less supported and drop out of a study. In the DEFINE trial retention was aided by keeping close 'virtual' contact with participants though nurse-led calls and automatic reminders, and contact routes in the event of problems. While not the focus of the qualitative interviews, transcript analysis has demonstrated that participants were supportive of the remote approach, noting flexibility in timing and reduced burden as strengths of the trial. The central point of contact, while remote, also ensured patients felt updated and in contact with the research team. Participant comments included "I felt very looked after by the trial delivery team" and the trial team "were on hand to help". One participant described the trial as a "great experience".

## Efficiency and cost savings

DEFINE's targeted social media advertising achieved rapid recruitment, with a direct advertising spend of only £880 ($1,148). This efficient approach yielded a total of 558 eligible participants after pre-screening, which compares favourably to a multiple site setup. The trial itself was also less expensive than a typical RCT, with a grant award value of £486,657 (£345,814 funder contribution). This included the cost of the interventions in DEFINE, with the Oto App being provided free of charge for the

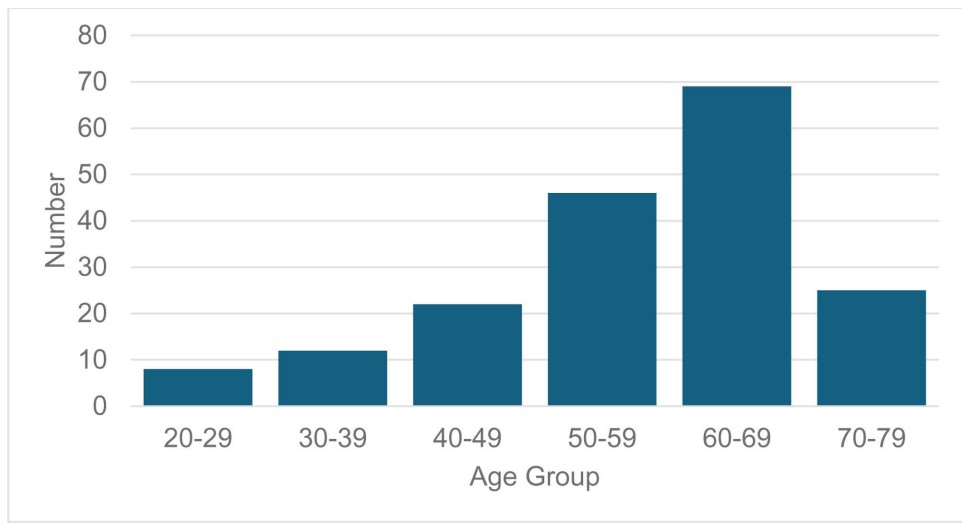

**Fig 5. Age of enrolled participants.**

study, as well as the CRO and investigator costs. Trials such as DEFINE eliminate costs associated with physical sites, such as set up and initiation visit costs, investigator salaries, data collection with local archiving, and participant travel costs.

In conventional trails requiring travel to a trial site, participants may need to front the cost of transport and may also lose income from lost wages and childcare costs. These costs can be a barrier to inclusion and may explain low participation amongst low-income groups.

## Other benefits of decentralised trials

One advantage of a central trial body directly interacting with patients is that the trial can rapidly respond to implement required changes, without having to filter protocol modifications down through sites. An example is seen by DEFINE; at the start of the study, participants were asked to complete baseline assessments after finishing the consent call within

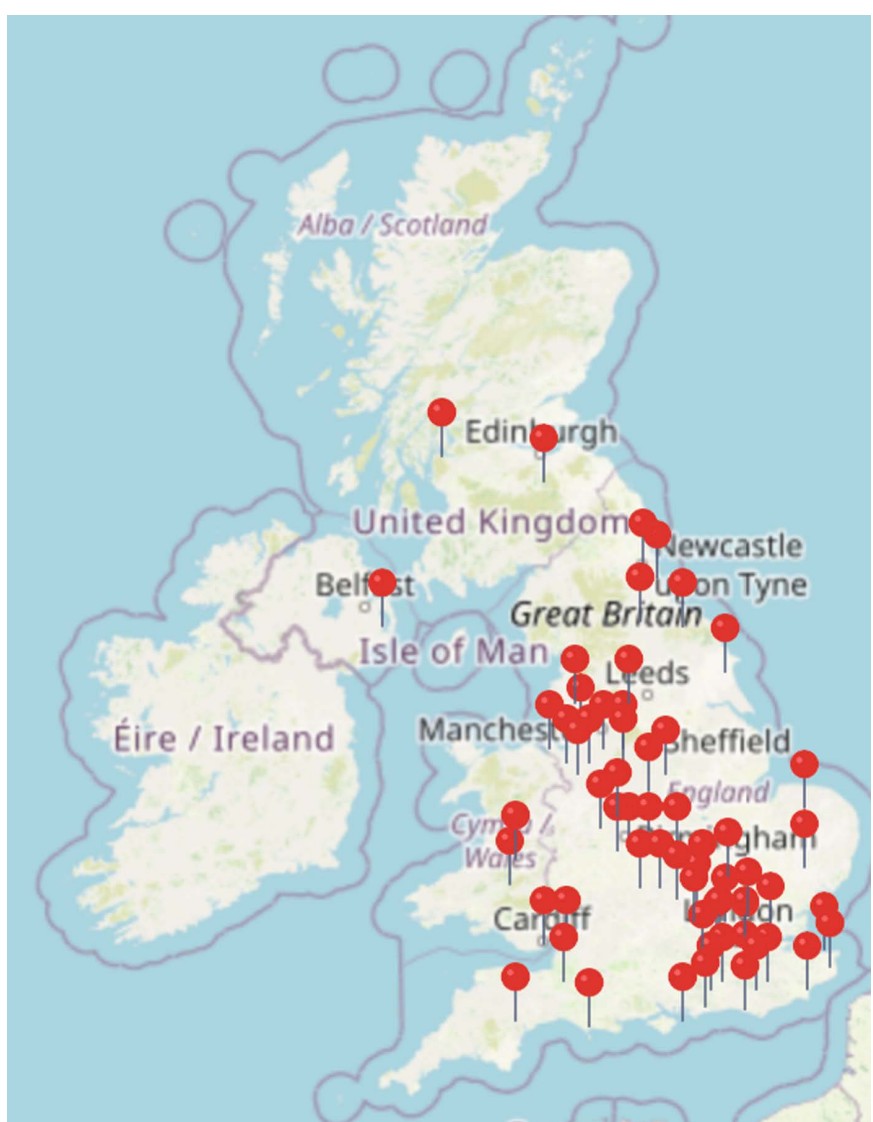

**Fig 6. Geographic distribution of participants in the Oto treatment arm.**

48 hours, but before starting treatment. Following initial recruitment, the centralised database enabled rapid identification that TFI data may be missing despite intervention start. The process could be immediately adapted with trial coordinators requesting that participants complete the baseline TFI during the consent call.

Decentralised trials potentially also represent a sustainable and environmentally friendly approach to trials. For DEFINE there was no participant or investigator travel, significantly reducing the carbon footprint of the study, with also reduced reliance on paper forms or physical data storage and archiving.

## Limitations of decentralised studies

Decentralised studies often heavily rely on technology, which can present difficulties. In DEFINE, some participants reported difficulties using the hearing test applications, with online document signing, as well as with accessing interventions. However, these issues were minor and easily resolved with the support of the central trial coordinators. User guides were produced and then adapted based on feedback to guiding participants through these technical steps. Overall, participants of all ages demonstrated good IT skills and wiliness to engage with technology.

Decentralised trials are increasingly well suited for hearing loss and tinnitus therapies. Accurate audiometry can now be performed at home either via a visiting health professional or with delivered calibrated equipment. Even smartphones paired with inexpensive earphones can provide accurate screening audiograms (as used in DEFINE) using free apps [10], though these still require refinement. Remote fitting of hearing aids has entered clinical practice, with uptake enhanced by the COVID-19 pandemic [17]. Complex interventions such as cochlear implants can also be remotely programmed by trained audiologists, and these devices further provide unique opportunities for high-frequency remote monitoring of devices and even cochlear neural health, a valuable asset for research [18].

New ethical considerations have been raised for decentralised trials, and these are still being explored in what is a new methodology [19]. Ensuring patients are who they claim to be, that they are suitable for inclusion, and can provide valid informed consent are all harder when assessed remotely. Identity checks can be built into the recruitment process, and videos and plain English documents can support consent, but most important is ensuring that the central CRO is staffed by experienced research staff, with appropriate oversight.

There will always be some interventions, whether complex to deliver, or high risk, that require face-to-face patient assessment, but by introducing aspects of decentralised trials into conventional designs, some of the benefits of efficiency and reduced patient burden can be seen.

## Conclusion

The fully decentralised approach adopted by DEFINE proved efficient and acceptable to patients. The success of this model suggests that it can be replicated for other conditions, especially where remote interventions and data capture are viable. The retention and adherence of participants recruited through social media compare favourably to those recruited in-person and locally. Decentralised trials have a role to play in providing cost effective, patient-centric research, potentially reaching some groups who historically struggle to engage with conventional trials. There are limitations to the application of the methodology, but hybrid approaches are likely to be of benefit in many clinical studies in the future.

## Supporting information

**S1 File. Control Arm Intervention Specification.**
(DOCX)

## Acknowledgments

We would like to thank the Independent Steering Group; Prof Simon Lloyd, Prof Nishchay Mehta and Mr Samir Mehta for their assistance with the study,

## Author contributions

**Conceptualization:** Jameel Muzaffar, Matthew E. Smith.

**Data curation:** Amy Moore, Olivia Dantonio, Luke Twelves, Michael Young.

**Formal analysis:** Joseph Salem, Jameel Muzaffar, Jan Multmeier.

**Investigation:** Dhiraj Sharma, Matthew E. Smith.

**Methodology:** Joseph Salem, Matthew E. Smith.

**Project administration:** Joseph Salem, Dhiraj Sharma, Jameel Muzaffar, Amy Moore, Olivia Dantonio, Luke Twelves, Emma Ogburn, Michael Young.

**Supervision:** Jameel Muzaffar, Matthew E. Smith.

**Writing – original draft:** Joseph Salem, Dhiraj Sharma, Jameel Muzaffar.

**Writing – review & editing:** Joseph Salem, Dhiraj Sharma, Jameel Muzaffar, Matthew E. Smith.

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
