## [Decision Letter · Decision Letter 0]

Dear Dr. Smith,

Thank you for submitting your manuscript to PLOS ONE. After careful consideration, we feel that it has merit but does not fully meet PLOS ONE’s publication criteria as it currently stands. Therefore, we invite you to submit a revised version of the manuscript that addresses the points raised during the review process.

We look forward to receiving your revised manuscript.

Kind regards,

Hantong Hu

Academic Editor

PLOS ONE

Journal Requirements:

2.  We note that your Data Availability Statement is currently as follows: “All relevant data are within the manuscript and its Supporting Information files.”

5. We note that Figure 1 and 2 in your submission contain copyrighted images. All PLOS content is published under the Creative Commons Attribution License (CC BY 4.0), which means that the manuscript, images, and Supporting Information files will be freely available online, and any third party is permitted to access, download, copy, distribute, and use these materials in any way, even commercially, with proper attribution. For more information, see our copyright guidelines: http://journals.plos.org/plosone/s/licenses-and-copyright.

A. You may seek permission from the original copyright holder of Figure 1 and 2 to publish the content specifically under the CC BY 4.0 license.

B. If you are unable to obtain permission from the original copyright holder to publish these figures under the CC BY 4.0 license or if the copyright holder’s requirements are incompatible with the CC BY 4.0 license, please either i) remove the figure or ii) supply a replacement figure that complies with the CC BY 4.0 license. Please check copyright information on all replacement figures and update the figure caption with source information. If applicable, please specify in the figure caption text when a figure is similar but not identical to the original image and is therefore for illustrative purposes only.

6. We note that Figure 6 in your submission contain map images which may be copyrighted. All PLOS content is published under the Creative Commons Attribution License (CC BY 4.0), which means that the manuscript, images, and Supporting Information files will be freely available online, and any third party is permitted to access, download, copy, distribute, and use these materials in any way, even commercially, with proper attribution. For these reasons, we cannot publish previously copyrighted maps or satellite images created using proprietary data, such as Google software (Google Maps, Street View, and Earth). For more information, see our copyright guidelines: http://journals.plos.org/plosone/s/licenses-and-copyright.

A. You may seek permission from the original copyright holder of Figure 6 to publish the content specifically under the CC BY 4.0 license. 

B. If you are unable to obtain permission from the original copyright holder to publish these figures under the CC BY 4.0 license or if the copyright holder’s requirements are incompatible with the CC BY 4.0 license, please either i) remove the figure or ii) supply a replacement figure that complies with the CC BY 4.0 license. Please check copyright information on all replacement figures and update the figure caption with source information. If applicable, please specify in the figure caption text when a figure is similar but not identical to the original image and is therefore for illustrative purposes only.

7. Please upload a copy of Supporting Information Figure/Table/etc. “supplementary material 2” which you refer to in your text on page 7.

Reviewers' comments:

Reviewer's Responses to Questions

**Comments to the Author**

1. Is the manuscript technically sound, and do the data support the conclusions?

Reviewer #1: Yes

Reviewer #2: No

2. Has the statistical analysis been performed appropriately and rigorously?

Reviewer #1: Yes

Reviewer #2: No

3. Have the authors made all data underlying the findings in their manuscript fully available?

Reviewer #1: Yes

Reviewer #2: Yes

4. Is the manuscript presented in an intelligible fashion and written in standard English?

Reviewer #1: Yes

Reviewer #2: Yes

Reviewer #1: Overall, a well structured manuscript with logical arguments for decentralized clinical trials, especially with hearing evaluation. Please clarify whether the trial use a stratified or unstratified randomization 1:1 ration. What statistical analytic methods were used to analyze the results. The manuscript will benefit with a more detailed discussion on the limitations of decentralized trials, including potential biases for research.

Reviewer #2: Dear authors,

This paper outlines the methodology of a randomized controlled trial for delivering remote hearing and tinnitus therapy. However, it closely resembles a previously published study protocol, "Digital Therapy for Improved Tinnitus Care Study (DEFINE): Protocol for a Randomised Controlled Trial."

The primary concern is that this paper does not present data on the treatment itself. Instead, it focuses on lessons learned from the study, which is more reflective of a report rather than a research paper.

**Do you want your identity to be public for this peer review?** For information about this choice, including consent withdrawal, please see our Privacy Policy

Reviewer #1: No

Reviewer #2: No

---

## [Author Response · Author response to Decision Letter 1]

29 Apr 2025

RESPONSE: We have checked that our manuscript and it follows the PLOS One Style requirements.

2. We note that your Data Availability Statement is currently as follows: “All relevant data are within the manuscript and its Supporting Information files.”

RESPONSE: We can confirm all relevant data is included. This is confirmed by both Reviewers who responded YES to the question “Have the authors made all data underlying the findings in their manuscript fully available?”.

RESPONSE: This has now been inserted into the Methods section as “). NHS ethical approval was granted by the West Midlands - Black Country Research Ethics Committee (REC Reference: 23/WM/0146) with all participants providing written consent to be involved in the study.”

RESPONSE: These have been added to the manuscript

5. We note that Figure 1 and 2 in your submission contain copyrighted images. All PLOS content is published under the Creative Commons Attribution License (CC BY 4.0), which means that the manuscript, images, and Supporting Information files will be freely available online, and any third party is permitted to access, download, copy, distribute, and use these materials in any way, even commercially, with proper attribution. For more information, see our copyright guidelines: http://journals.plos.org/plosone/s/licenses-and-copyright.

A. You may seek permission from the original copyright holder of Figure 1 and 2 to publish the content specifically under the CC BY 4.0 license.

B. If you are unable to obtain permission from the original copyright holder to publish these figures under the CC BY 4.0 license or if the copyright holder’s requirements are incompatible with the CC BY 4.0 license, please either i) remove the figure or ii) supply a replacement figure that complies with the CC BY 4.0 license. Please check copyright information on all replacement figures and update the figure caption with source information. If applicable, please specify in the figure caption text when a figure is similar but not identical to the original image and is therefore for illustrative purposes only.

RESPONSE: We have requested the permissions from the relevant parties and attached the forms to our submission. The relevant copyright caption has also been applied.

6. We note that Figure 6 in your submission contain map images which may be copyrighted. All PLOS content is published under the Creative Commons Attribution License (CC BY 4.0), which means that the manuscript, images, and Supporting Information files will be freely available online, and any third party is permitted to access, download, copy, distribute, and use these materials in any way, even commercially, with proper attribution. For these reasons, we cannot publish previously copyrighted maps or satellite images created using proprietary data, such as Google software (Google Maps, Street View, and Earth). For more information, see our copyright guidelines: http://journals.plos.org/plosone/s/licenses-and-copyright.

A. You may seek permission from the original copyright holder of Figure 6 to publish the content specifically under the CC BY 4.0 license.

B. If you are unable to obtain permission from the original copyright holder to publish these figures under the CC BY 4.0 license or if the copyright holder’s requirements are incompatible with the CC BY 4.0 license, please either i) remove the figure or ii) supply a replacement figure that complies with the CC BY 4.0 license. Please check copyright information on all replacement figures and update the figure caption with source information. If applicable, please specify in the figure caption text when a figure is similar but not identical to the original image and is therefore for illustrative purposes only.

RESPONSE: We have replaced our GoogleMaps image with one of the open source maps suggested above.

7. Please upload a copy of Supporting Information Figure/Table/etc. “supplementary material 2” which you refer to in your text on page 7.

RESPONSE: Our supplementary material 2 has been renamed supplementary material 1 and added to our file.

Reviewers' comments:

Reviewer's Responses to Questions

Comments to the Author

1. Is the manuscript technically sound, and do the data support the conclusions?

Reviewer #1: Yes

Reviewer #2: No

RESPONSE: The Reviewer incorrectly states this paper “outlines the methodology of a randomized controlled trial for delivering remote hearing and tinnitus therapy”. However, this is not correct as the paper outlines the learnings of delivering the first decentralised control trial for an ENT research topic. The results are presented showing new data, not only a methodology, and these results are discussed later in the paper as part of the wider literature.

2. Has the statistical analysis been performed appropriately and rigorously?

Reviewer #1: Yes

Reviewer #2: No

RESPONSE: The appropriate analysis of the data is presented in our results section.

3. Have the authors made all data underlying the findings in their manuscript fully available?

Reviewer #1: Yes

Reviewer #2: Yes

RESPONSE: No comment

4. Is the manuscript presented in an intelligible fashion and written in standard English?

Reviewer #1: Yes

Reviewer #2: Yes

RESPONSE: No comment

5. Review Comments to the Author

Reviewer #1: Overall, a well structured manuscript with logical arguments for decentralized clinical trials, especially with hearing evaluation. Please clarify whether the trial use a stratified or unstratified randomization 1:1 ration. What statistical analytic methods were used to analyze the results. The manuscript will benefit with a more detailed discussion on the limitations of decentralized trials, including potential biases for research.

RESPONSE: We appreciate the comments made by Reviewer 1.

We can confirm stratified randomisation 1:1 ratio was implemented, and this has been integrated into the methodology in the main text.

Regarding the statistical analysis, our data is presented as percentages and proportions in our results section. Complex statistical analysis was not required as part of the data analysis regarding the study feasibility and role of decentralised randomised control trials. The learnings from the data is useful to the medical community.

We rebut the comment regarding “The manuscript will benefit with a more detailed discussion on the limitations of decentralized trials” as there is a sizeable section outlining the Limitations of decentralised studies which is labelled as such on line 369.

Reviewer #2: Dear authors,

This paper outlines the methodology of a randomized controlled trial for delivering remote hearing and tinnitus therapy. However, it closely resembles a previously published study protocol, "Digital Therapy for Improved Tinnitus Care Study (DEFINE): Protocol for a Randomised Controlled Trial."

The primary concern is that this paper does not present data on the treatment itself. Instead, it focuses on lessons learned from the study, which is more reflective of a report rather than a research paper.

RESPONSE: We appreciate the comments made by Reviewer 2 however disagree regarding the similarity of our paper with the protocol study quoted and that the paper is a reflective report.

The study protocol that has been previously published outlines what the study had planned to complete. This paper provides new information on study feasibility of a new methodology of decentralised randomised control trials in a field where it has not been previously used. It provides novel data that provides benefit to the medical community in addition to that of the published protocol.

We also disagree that the paper does not present data to warrant being a research paper. The paper outlines the lessons learnt from a novel methodology and are not a reflective account of the study. Data on the outcomes of the study methodology is provided and demonstrates the feasibility of the decentralised randomised control trials.

---

## [Editor Report · Decision Letter 1]

Decentralised trials for hearing and tinnitus therapies: Lessons from the Digital thErapy For Improved tiNnitus carE (DEFINE) Randomised Controlled Trial

PONE-D-25-04773R1

Dear Dr. Smith,

We’re pleased to inform you that your manuscript has been judged scientifically suitable for publication and will be formally accepted for publication once it meets all outstanding technical requirements.

Kind regards,

Hantong Hu

Academic Editor

PLOS ONE
---

## [Editor Report · Acceptance letter]

PONE-D-25-04773R1

PLOS ONE

Dear Dr. Smith,

I'm pleased to inform you that your manuscript has been deemed suitable for publication in PLOS ONE. Congratulations! Your manuscript is now being handed over to our production team.

Kind regards,

on behalf of

Dr. Hantong Hu

Academic Editor

PLOS ONE